# Relationship between Nitrogen Dynamics and Key Microbial Nitrogen-Cycling Genes in an Intensive Freshwater Aquaculture Pond

**DOI:** 10.3390/microorganisms12020266

**Published:** 2024-01-26

**Authors:** Yifeng Yan, Junbo Zhou, Chenghao Du, Qian Yang, Jinhe Huang, Zhaolei Wang, Jun Xu, Min Zhang

**Affiliations:** 1Hubei Provincial Engineering Laboratory for Pond Aquaculture, Engineering Research Center of Green Development for Conventional Aquatic Biological Industry in the Yangtze River Economic Belt, College of Fisheries, Huazhong Agricultural University, Wuhan 430070, China; yanyifeng2@163.com (Y.Y.); 18523377746@163.com (J.Z.); dch128@outlook.com (C.D.); yang123321@foxmail.com (Q.Y.); huangjinhe@webmail.hzau.edu.cn (J.H.); w13483003139@163.com (Z.W.); 2Institute of Hydrobiology, Chinese Academy of Sciences, Wuhan 430072, China; xujun@ihb.ac.cn

**Keywords:** nitrogen-cycling genes, freshwater fishponds, sediment–water interface, suspended particulate matter, nitrogen transformation processes

## Abstract

Intensive aquaculture in high-density hybrid snakehead [*Channa maculata* (♀) × *Channa argus* (♂)] fishponds can lead to toxic conditions for fish. This study investigated nitrogen migration and transformation in these fishponds during different cultivation periods. Using qPCR technology, we analyzed the abundance variation of nitrogen-cycling microorganisms in water and sediment to reveal the nitrogen metabolism characteristics of hybrid snakehead fishponds. The results showed that fish biomass significantly impacts suspended particulate matter (SPM) flux. At the sediment–water interface, inorganic nitrogen fluxes showed predominant NO_3_^−^-N absorption by sediments and NH_4_^+^-N and NO_2_^−^-N release, especially in later cultivation stages. Sediments were rich in *nirS* and AMX 16S rRNA genes (ranging from 4.04 × 10^9^ to 1.01 × 10^10^ and 1.19 × 10^8^ to 2.62 × 10^8^ copies/g, respectively) with *nirS*-type denitrifiers potentially dominating the denitrification process. Ammonia-oxidizing bacteria (AOB) were found to dominate the ammonia oxidation process over ammonia-oxidizing archaea (AOA) in both water and sediment. Redundancy analysis revealed a positive correlation between SPM flux, Chlorophyll a (Chl-a), and denitrification genes in the water, and between nitrogen-cycling genes and NH_4_^+^/NO_2_^−^ fluxes at the interface. These findings provide a scientific basis for nitrogen control in hybrid snakehead fishponds.

## 1. Introduction

Aquaculture holds a crucial position in meeting the protein demands of the swiftly expanding world population, with particular significance in China [1]. However, intensive aquaculture, while resulting in high production, leads to the release of nutrient-rich aquaculture wastewater, increasing the nutrient load in water bodies and potentially causing eutrophication [2]. Although the use of recirculating aquaculture systems (RAS) can mitigate nutrient release in fish farming, RAS systems have not been widely adopted due to their higher water treatment costs [3]. Consequently, intensive pond aquaculture will continue to be important in developing countries.

In China, hybrid snakehead [*Channa maculata* (♀) × *Channa argus* (♂)] is one of the species adapted to intensive pond aquaculture. Under an intensive farming model with a six-month culture cycle, production can reach 90,000–112,500 kg/ha, providing significant yields and economic value [4]. Given the implementation of wastewater treatment policies, intensive aquaculture models are expected to continue in China. However, nitrogen pollution from intensive aquaculture can have toxic effects on aquaculture animals, particularly the significant accumulation of ammonia nitrogen and nitrite [5].

In the context of aquaculture water quality management, researchers have directed their attention toward the key factors influencing nitrogen cycling, namely microorganisms, with the aim of mitigating nitrogen pollution. They have studied the abundance, community structure, and reaction efficiency of nitrifying, denitrifying, and anaerobic ammonia-oxidizing (anammox) microorganisms in pond sediments. AOA *amoA* is more abundant than AOB *amoA*, and AOA *amoA* is influenced by factors such as dissolved oxygen (DO) and total organic carbon [6,7]. However, some studies have shown that AOB *amoA* can be more abundant than AOA *amoA*, which may be due to differences in aquaculture species and environmental conditions [8,9]. In addition, both AOA and AOB are susceptible to light inhibition, with AOA being more sensitive. Chen et al. [10] discovered that increasing the C/N ratio leads to an elevation in the abundance of denitrification genes and an increase in nitrification rates. In zero water exchange ponds, denitrification rates are positively correlated with sediment nitrate concentrations. Environmental factors primarily influence denitrification rates rather than community structure [11]. Anammox is mainly influenced by factors such as organic carbon content, nitrite, and ammonia concentrations [12,13].

Beyond analyzing microbial characteristics associated with nitrogen cycling, exploring how nutrients are transported and transformed within pond ecosystems can deepen our knowledge of nitrogen regulatory processes. The aquatic environment is where fish directly reside, and sediments have a significant impact on the nutrient status of water [14]. Depending on environmental conditions, sediments can act as sources or sinks for nutrients, releasing or absorbing them into the water, respectively [15]. Petranich et al. [16] found that nutrient accumulation in aquaculture ponds could promote intense remineralization at the sediment–water interface. It has been observed that disturbance by carp in sediments can enhance nitrogen removal [17]. Furthermore, suspended particulate matter (SPM) plays a crucial role as a link between water and sediment, influencing nutrient exchange and geochemical processes in ponds [18]. An increase in suspended particles may lead to nutrient accumulation of nutrients in sediments and affect denitrification processes in the water [5,19].

However, many studies have primarily focused on nitrogen-related functional genes in sediments. Rarely has research combined studies of both water columns and sediments, and considered the implications for nutrient transport and transformation in ponds. This comprehensive approach is essential to achieve a more thorough understanding of nitrogen-cycling processes in freshwater fishponds. Therefore, the objectives of this study were (1) to describe the spatial patterns of nutrient concentrations within the ponds, (2) to quantify the temporal distribution of N cycling genes (AOA *amoA*, AOB *amoA*, *narG*, *nirS*, *nirK*, and anammox (AMX) 16S rRNA) in water and sediment, and (3) to investigate the relationships between nutrient concentrations and the abundance of nitrogen-cycling genes in both water and sediment. Finally, this study aims to identify the key factors that may influence the management of fish farms and to propose feasible recommendations to improve the aquatic environment of such fishponds.

## 2. Materials and Methods

### 2.1. Site Description

Three freshwater ponds, each containing hybrid snakehead [*Channa maculata* (♀) × *Channa argus* (♂)] for commercial use, were investigated. These ponds are located in Sanjiao Town (113.41° E, 22.65° N), Zhongshan City, China. Each sampling pond had a surface area of approximately 5000 m^2^ and a depth ranging from 2 to 2.5 m. These ponds were independent of each other, received water from a nearby stream, and were equipped with five aerators to enhance dissolved oxygen levels. Shade nets were placed over the pond in July for the purpose of shading, followed by the addition of white plastic film in October for insulation. The estimated fish densities for pond A1, pond A2, and pond A3 were 96,297, 102,031, and 103,095 individuals per pond, respectively. Fish were fed with commercial floating pellets containing 40% total protein, 5% fat, and 0.8% total phosphorus (Haid Company, Zhuhai City, China). The monthly ration (Kg/month) was determined based on fish biomass and feeding responses. The total feed quantities for pond A1, pond A2, and pond A3 were 69,232 kg, 98,989 kg, and 97,186 kg, respectively.

### 2.2. Sample Collection

Mid-monthly sampling was executed from June to November in the year 2022. Surface water samples were collected from five points 0.5 m below the water surface using a water sampler and mixed together. They were then placed in sterile plastic bottles and stored in an ice cooler before being transported to the laboratory. One part of the samples was left unfiltered, while the other part was filtered through 0.45 μm membranes and then both sets of samples were frozen at −20 °C for analysis; the rest of the water samples for molecular analysis and Chlorophyll a (Chl-a) determination was filtered through 0.22/0.45 μm membranes. The 0.22 μm filters were stored at −80 °C until DNA extraction, whereas the 0.45 μm filters were kept at −20 °C until the Chl-a concentration measurements were conducted.

Overlying water samples were collected using an in situ benthic chamber [20], constructed from Plexiglas and measuring 20 × 20 × 20 cm. The chamber was carefully placed at the central bottom of each pond, where it was allowed to settle for 30 min. Water samples were then extracted from the chamber using a 100 mL plastic syringe attached to a silicone hose. To assess the inorganic nitrogen fluxes, the device was incubated undisturbed for 24 h, after which another overlying water sample was collected. All samples were filtered through 0.45 μm membranes and preserved at −20 °C for later analysis.

SPM were collected using sediment traps at five points in each pond bottom for 48 h. The sediment traps were made from cylindrical plastic pipe with a diameter of 9.2 cm and a height of 16 cm. Traps were placed at 10 cm above the sediment. After allowing the collected SPM samples to settle for a duration of 2 h, the supernatant was carefully removed. The samples were then dried at 60 °C to a constant weight, then ground and passed through a 150 μm sieve. Finally, they were stored in a desiccator for analysis.

Sediment samples were collected at the same locations of sedimentary particles from 0 to 10 cm depth using a Peterson grab sampler. An equal amount of the sediment was taken in each pond and mixed together. The collected sediment was immediately placed in a sterile plastic bag. It was then separated into two parts. The first part was dried at 60 °C, then ground, passed through a 150 μm sieve, and stored in a desiccator for physicochemical analysis. The second part was frozen at −80 °C for DNA extraction.

### 2.3. Physico-Chemical Parameter Analysis

Surface and overlying water temperature (T), pH, and dissolved oxygen (DO) were measured using a YSI Pro1020 water quality meter (YSI Inc., Yellow Springs, OH, USA), equipped with electrodes for precise measurement of these parameters (https://www.ysi.com/pro1020, accessed on 21 January 2024). Water transparency was measured with a Secchi disk (SD). Chlorophyll a (Chl-a) was determined with the hot ethanol method [21]. The analysis of nitrate (NO_3_^−^-N), ammonium (NH_4_^+^-N), nitrite (NO_2_^−^-N), phosphorus (PO_4_^3−^-P), total phosphorus (TP_w_), and total nitrogen (TN_w_) concentrations in the water column was conducted using the UV-1900i UV-VIS spectrophotometer (SHIMADZU Corporation, Kyoto, Japan). These analyses were performed according to the protocols described in the Monitoring Analysis Method of Water and Wastewater [22]. The same instrument was also used for determining the total nitrogen (TN_S_) and total phosphorus (TP_S_) in sediments and total nitrogen (TN_SPM_) and total phosphorus (TP_SPM_) in SPM by high-temperature digestion with alkaline potassium persulfate [23].

### 2.4. Inorganic Nitrogen Fluxes and SPM Flux Calculation

Inorganic nitrogen fluxes across the sediment–water interface were measured using an in situ benthic chamber. Positive values indicate efflux, whereas negative values represent influx or scavenging from the water column into the sediment particles. Inorganic nitrogen fluxes were calculated using the following equation:$$F=\frac{∆C\cdot V}{A\cdot ∆t}$$
where *F* (mg m^−2^ d^−1^) represents inorganic nitrogen fluxes; *V* (m^3^) is the volume of the chamber; *A* (m^2^) is the bottom area of the chamber; Δ*t* (d) is the incubation duration; and Δ*C* (mg/L) is the change in the concentrations of ammonia, nitrite, and nitrate before and after incubation.

SPM flux was calculated according to the following formula:$$f=\frac{m}{h\cdot a}\;f_{x}=f\cdot C_{x}$$
where f (g m^−2^ d^−1^) stands for SPM flux; m (g) is the dry weight of sedimentary particles; h (d) denotes the collection duration; a (m^2^) is the area of the sampling funnel; $f_{x}$ (mg m^−2^ d^−1^) represents SPM flux of nutrient element x; $C_{x}$ (mg/g) is the content of element x.

### 2.5. Total DNA Extraction and Quantitative PCR

DNA from water and sediment samples was extracted from 0.22 μm membrane (filtered 50 mL surface water) and 0.3 g sediment (wet weight) using the Water DNA Isolation Kit (Foregene, Chengdu, China) and ALFA-Soil DNA Extraction Mini Kit (Findrop, Guangzhou, China) according to the manufacturer’s instructions. The quality of the extracted DNA was verified by electrophoresis in 1% agarose, and the DNA concentration was determined using a Nanodrop 2000 spectrophotometer (Thermo Fisher Scientific, Waltham, MA, USA). The DNA samples were stored at −20 °C for subsequent molecular analyses.

Quantitative real-time polymerase chain reaction (qPCR) assay was applied to determine the number of copies of bacterial and Crenarchaeota 16S rRNA gene and N-cycling genes (AOA *amoA*, AOB *amoA*, *narG*, *nirS*, *nirK*, AMX 16S rRNA). A qTOWER^3^ real-time PCR system (Analytikjena, Jena, Germany) was used to perform qPCR measurement. Quantification was based on the fluorescent dye SYBR-Green using 2X SYBR Green Abstart PCR Mix (Sangon, Shanghai, China). The PCR reaction mixture (20 μL) included 2X SYBR Green Abstart PCR Mix (10 μL), forward and reverse primers (0.4 μL), sterile double-distilled H_2_O (to 20 μL), and DNA template (2 μL). Optimized qPCR thermal profiles and the primers used for target genes and standard curves are listed in Table 1 and Table 2. Standard curves were obtained by serial dilution (10^−3^ to 10^−8^) of the plasmids containing target gene fragments. No template control and standard curve template DNA was amplified in triplicate, on the same plate as the environmental samples. Amplification efficiencies ranged from 76% to 107%, with R^2^ > 0.99.

### 2.6. Statistical Analysis

Data are expressed as mean ± standard deviation. The qPCR data were analyzed using the Analytikjena qPCRsoft 4.1 (Analytikjena, Jena, Germany). Quantitative analysis results were expressed as gene copies mL^−1^ of water and copies g^−1^ of wet-weight sediment. Gene abundances were calculated based on standard curves and then converted to gene copy numbers assuming 100% DNA extraction efficiency.

The Spearman correlation analysis between environmental parameters was performed using R (version 3.4.1). A one-way analysis of variance (ANOVA) was utilized to assess the differences in physicochemical indicators at various sampling times. A ridge regression analysis was conducted to evaluate the relative contributions of influencing variables to the SPM flux using SPSS 26.0 software. Redundancy analysis (RDA) was performed using CANOCO 5.0 (Biometris Inc., Wageningen, The Netherlands) to reveal the relationship between functional genes and a range of environmental factors.

## 3. Results

### 3.1. Physicochemical Characteristics of the Aquaculture Ponds

Figure 1 and Table 3 show the physicochemical characteristics of the surface water, sediments, overlying water, and SPM. T, pH, DO and SD values of the surface water ranged from 28.8 to 31.6 °C, 7.3 to 8.3, 3.1 to 4.8 mg/L, and 17 to 37 cm, respectively. Due to the use of shade nets in July and white plastic film in October, there was relatively little variation in pond water temperature during the cultivation period. Both pH and SD exhibited a simultaneous decreasing trend, while Chl-a showed an increasing trend, ranging from 50.7 to 411.73 mg/m^3^.

NH_4_^+^-N and NO_2_^−^-N in surface water both peaked in September, ranging from 2.07 to 9.87 and 0.59 to 5.97 mg/L, respectively. NO_3_^−^-N and TN_W_ showed a gradual upward trend, ranging from 2.6 to 24.34 and 10.38 to 36.94 mg/L, respectively. The overlying water concentrations of NH_4_^+^-N, NO_2_^—^N, and NO_3_^−^-N showed a consistent pattern of variation. PO_4_^−^-P and TP_W_ reached the maximum in September and October (2.78 and 3.98 mg/L) and the minimum in November (1.01 and 2.61 mg/L).

The SPM flux showed a significant increase, rising from 239.97 g m^−2^ d^−1^ in June to 2398.96 g m^−2^ d^−1^ in November. The SPM fluxes of TN (TN_SPM_ flux) and TP (TP_SPM_ flux) also exhibited an increasing trend, ranging from 1.93 to 19.18 and 2.71 to 15.59 g m^−2^ d^−1^, respectively. TN_S_ concentrations peaked in November with values ranging from 3.42 to 6.00 mg/g, while TP_S_ concentrations peaked in October with values ranging from 3.99 to 6.08 mg/g, both indicating a gradual increase.

### 3.2. Inorganic Nitrogen Fluxes across the Sediment–Water Interface

Variations of inorganic nitrogen fluxes across the sediment–water interface are shown in Figure 2. Positive values indicate effluxes, while negative values are representative of influxes. The results showed that the fluxes of NH_4_^+^-N, NO_2_^−^-N, and NO_3_^−^-N ranged from −310.79 to 288.55, −57.45 to 190.56, and −569.46 to 450.67 mg m^−2^ d^−1^, respectively. NH_4_^+^-N and NO_2_^−^-N showed influxes in the month of June, indicating that these nutrients in the overlying water are taken in by the sediments at the beginning of the culture, but showed effluxes in the following months (NH_4_^+^-N changed in July, NO_2_^−^-N in September). NO_3_^−^-N generally showed influxes, with the exception of August (450.67 mg m^−2^ d^−1^), indicating that NO_3_^−^-N was absorbed by the sediment from the water column in most months.

### 3.3. Relationships between Environmental Factors in Fishponds

Based on the results of the Spearman correlation analysis performed between water parameters and sediment characteristics (Figure 3), NO_3_^−^-N flux displayed no significant correlations with other environmental factors. Conversely, NO_2_^−^-N, NO_3_^−^-N, TN_W_, SPM flux, TN_SPM_ flux, TP_SPM_ flux, feed ration, NH_4_^+^-N flux, NO_2_^−^-N flux, TN_S_, TP_S_, and Chl-a exhibited significant positive correlations, with the majority showing negative correlations with DO, pH, and SD.

Using feed ration, Chl-a, and fish biomass as independent variables, and SPM flux as the dependent variable, a ridge regression analysis with a K value of 0.13 was performed. The model equation is as follows: SPM flux = −150.768 + 0.009 × feed ration + 0.689 × Chl-a + 0.309 × fish biomass. The results indicate that fish biomass (t = 8.342, *p* < 0.01) has a significant positive effect on SPM flux. The relationships between Chl-a (t = 1.130, *p* = 0.264) and feed ration (t = 0.769, *p* = 0.445) with SPM flux are not significant, suggesting that their explanatory power for SPM flux is relatively weak (Table 4).

### 3.4. Variations in the Abundance of Functional Genes in Water and Sediment

The variation in the abundance of nitrogen cycle genes at different culture stages is shown in Figure 4. PCR analysis successfully detected the presence of the Crenarchaeota 16S rRNA gene and the AOA *amoA* gene. However, due to their low concentrations, precise quantification was not possible. In the sediment samples, the abundance of AOB *amoA* (ranging from 1.2 × 10^7^ to 2.6 × 10^7^ copies/g) was about one order of magnitude higher than the abundance of AOA *amoA* (ranging from 9.1 × 10^5^ to 1.7 × 10^6^ copies/g), indicating that AOB was predominant. The abundance of *nirS* exceeded that of *nirK* in all water and sediment samples (approximately *nirS* is 10 to 100 times more abundant than *nirK*). The range of variation of *narG* in water was from 2.9 × 10^4^ to 6.7 × 10^4^ copies/mL, while in sediment it was from 1.7 × 10^7^ to. 4.0 × 10^7^ copies/g. The AMX 16S rRNA gene is sparsely distributed in the water column (ranging from 2.3 × 10^3^ to 5.2 × 10^3^ copies/mL), but abundantly distributed in the sediment (ranging from 1.2 × 10^8^ to 2.6 × 10^8^ copies/g).

In the water column, the average abundance of the bacterial 16S rRNA gene was found to be 7.91 × 10^7^ copies/mL, with the *nirS* gene being the most abundant nitrogen cycle gene with an average of 1.25 × 10^6^ copies/mL. In the sediment, the average abundance of the bacterial 16S rRNA gene was 7.63 × 10^10^ copies/g, with Crenarchaeota averaging 9.0 × 10^8^ copies/g. The *nirS* gene was the most abundant represented nitrogen cycle gene in the sediment, with an average abundance of 7.03 × 10^9^ copies/g.

### 3.5. Relationships between Functional Gene Abundance and Environmental Factors

The RDA analysis correlating environmental variables with gene abundances in the water column indicated that the first two RDA axes explained 58.64% of the variance (Figure 5A). The Monte Carlo permutation test on the RDA data identified SPM flux (21.2% explained variance), TN (11.9% explained variance), SD (9.5% explained variance), and NO_3_^−^-N (4.7% explained variance) as the main environmental factors affecting gene distribution (Appendix A). AMX 16S rRNA and AOB *amoA* exhibited negative correlations with T, PO_4_^3−^-P, TP_W_, NH_4_^+^-N, and NO_2_^−^-N. On the other hand, *nirS*, *nirK*, and bacterial 16S rRNA showed positive associations with SPM flux, TN_W_, and Chl-a. Furthermore, *narG* showed positive correlations with TP, NH_4_^+^-N, NO_2_^−^-N, and Chl-a. AOB *amoA* and AMX 16S rRNA abundances were significantly positively correlated, while both showed negative correlations with *narG*. Bacterial 16S rRNA, *nirS*, and *nirK* were positively correlated.

The RDA analysis of environmental variables and gene abundances in the sediment revealed that the first two RDA axes accounted for 63.5% of the variance (Figure 5B). The Monte Carlo permutation test results for this RDA showed that NH_4_^+^-N flux (20.8% explained variance), NO_3_^−^-N flux (12% explained variance), TN_SPM_ flux (9.3% explained variance), and O-NO_3_^−^-N) (10.3% explained variance) were the main factors influencing gene distribution (Appendix A). AOA *amoA* and AOB *amoA* exhibited positive correlations with TN_SPM_ flux, TP_SPM_ flux, TN_S_, TP_S_, NH_4_^+^-N flux, NO_2_^−^-N flux, O-NH_4_^+^-N, and O-NO_3_^−^-N. *nirS*, *narG*, and bacterial 16S rRNA showed positive correlations with NH_4_^+^-N flux, NO_2_^−^-N flux, TP_S_, and O-NO_2_^−^-N. AMX 16S rRNA and *nirK* displayed positive correlations with O-NO_2_^−^-N, NH_4_^+^-N flux, and NO_2_^−^-N flux, but negative correlations with NO_3_^−^-N flux. AOA *amoA* and AOB *amoA* were positively correlated, as were AMX 16S rRNA and *nirK*. AOA *amoA* exhibited positive correlations with bacterial 16S rRNA, *nirS*, and *narG*.

## 4. Discussion

### 4.1. Distribution Characteristics of Nitrogen in Different Culture Periods

In aquaculture, the external addition of organic matter to ponds escalates with increasing feed input. Feed ration shows a positive correlation with TN_S_ (r = 0.77, *p* < 0.05). Fish excreta and a proportion of feed residues settle in the sediment, where nitrogen is degraded and deposited, leading to an increase in TN_S_ [24].

Particles in the water column ultimately enter the sediment through processes of suspension, adsorption, and settling, with the SPM acting as a critical link between suspended particles in the water column and the sediment [18]. Therefore, suspended particle concentrations increase as SPM flux increases. In this study, the increasing trend of SPM flux corresponds to the observed increase in the white seabream pond [25], with both exhibiting significant increases in the later culture stages (Figure 1). As the SPM in the fishpond is mainly derived from phytoplankton, feces, resuspended sediment, and feed ration [26,27], a ridge regression analysis was conducted on SPM flux in relation to Chl-a, feed ration, and fish biomass. Ridge regression analysis within our model framework indicates a significant effect of fish biomass on SPM flux, in contrast to the non-significant role of Chl-a and feed ration (Table 3). These results are consistent with previous research highlighting the integral role of biotic disturbances in modulating SPM dynamics in aquaculture systems [28,29]. Considering that feeding practices in aquaculture are adjusted based on the feeding behavior of the fish, resulting in a significant reduction of uneaten feed, the role of ration in influencing SPM flux may be mainly manifested through fish excretion. Therefore, the significant increase in SPM flux in the later stages of aquaculture may be mainly due to sediment resuspension enhanced by disturbance and fish excretion.

SPM flux shows a positive correlation with TN_W_ (r = 0.76, *p* < 0.05). Increases in SPM flux are indicative of increased organic matter content in the water body, resulting in increased levels of TN_W_. The presence of abundant TN_W_ facilitates nitrification processes, culminating in the accumulation of NO_3_^−^-N (Table 3). Algal growth is regulated by the TN:TP ratio, with nitrogen limitation occurring when the ratio is below 14 [30]. The escalation of TN_W_ concentrations alleviates the nitrogen limitation on algal growth, resulting in a noticeable increase in Chl-a. An increase in both Chl-a and SPM fluxes has an inverse effect on water transparency (Figure 3).

Fish activity creates disturbances that enhance the penetration of dissolved oxygen into the surface sediments, thereby promoting the mineralization of organic matter [31,32]. Rising temperatures enhance the metabolic functions of mineralizing microbes, thereby increasing oxygen demand and consequently decreasing DO levels in the overlying water during the aquaculture process (Table 3). At the same time, temperature increase also enhances the production and release of NH_4_^+^-N in the sediment. Similar to the findings of Seiki, T. [33], there is a positive correlation between T and NH_4_^+^-N flux (r = 0.38, *p* < 0.05). Elevated temperatures promote the mineralization of organic matter in the sediment, potentially resulting in excess NH_4_^+^-N release to the overlying water, exceeding the nitrification requirements. Furthermore, increasing fish biomass intensifies sediment resuspension, thereby increasing the frequency of sediment–water exchange and leading to the transfer of NH_4_^+^-N from sediment pore water to the overlying water [34].

### 4.2. Distribution and Variation of Nitrogen Cycle Genes in Water and Sediment

In the water column, ammonia-oxidizing archaea (AOA) are present in relatively low abundance compared to ammonia-oxidizing bacteria (AOB). The dominance of AOB may be due to their lower sensitivity to the photoinhibitory effects that affect AOA [6,35]. Specifically, AOB *amoA* concentrations in the water decrease from July to October, likely due to light exposure and algal growth inhibitory effects [36]. In sediments, AOB *amoA* are an order of magnitude more abundant than AOA *amoA*, reflecting trends seen in mandarin fishponds, in contrast to carp fishponds [8,37]. In low oxygen zones, AOA appears to have a competitive advantage [38]. The high-density, benthic habitat of the hybrid snakehead in our study likely causes increased sediment disturbance and oxygenation, favoring AOB.

The spatial distribution of the AMX 16S rRNA gene (Figure 4) reveals its limited presence in surface water but significant abundance in sediment. This pattern suggests that anammox bacteria are adapted to sedimentary environments where they play an important role in nitrogen removal. The abundance of the AMX 16S rRNA gene in the sediment of hybrid snakehead ponds is significantly higher than in estuarine and silver carp pond environments [39,40], possibly due to the high NH_4_^+^-N and NO_2_^−^-N levels, which provide substrates for anaerobic ammonium oxidation processes [41]. In addition, the sparse representation of anammox bacteria in the water column may be due to competitive interactions with denitrifying bacteria on suspended particulate matter.

Denitrifying bacteria, which are distributed throughout both the water column and sediment, show a significantly higher abundance than anammox bacteria (Figure 4). This predominance suggests a critical role for denitrifying bacteria in the nitrogen removal process of the pond. Our results also indicate a significantly higher prevalence of the *nirS* gene compared to the *nirK* gene, a finding consistent with different aquatic environments [10,42]. The abundance of the *nirS* gene in sediments exceeds that found in estuaries [43], shrimp ponds [9], and zero water exchange ponds [11], which may be regulated by the inorganic nitrogen levels in the sedimentary environments [13].

### 4.3. Nitrogen Cycle Process Mediated by Functional Gene and Environmental Factors in Fishpond

The abundance of *nirS* and *nirK* genes was found to be positively correlated with the SPM flux (Figure 5A), indicating a favorable condition for denitrifying bacteria within the SPM of the fishpond. This finding is consistent with observations from different aquatic environments such as the Yellow River [44], Poyang Lake [45], and Hangzhou Bay [42], where an increase in suspended particle concentration was associated with increased denitrification activity. The preference of denitrifying bacteria for anaerobic conditions, often found in suspended particle microenvironments, supports the occurrence of denitrification and coupled nitrification–denitrification processes [19].

In addition, our study reveals a significant positive correlation between the abundance of *narG*, *nirS*, and *nirK* genes and Chl-a concentration (Figure 5A). This suggests that algal aggregates, which are common during bloom events, provide a favorable environment for denitrifying bacteria. This phenomenon is supported by similar findings in lake studies during algal blooms [46,47,48]. The anaerobic conditions promoted by gel-like substances from algal aggregates facilitate these processes [49]. However, it is worth noting that ammonium assimilated by algae can be remineralized back into ammonium during the decay phase [50], potentially explaining the ammonia nitrogen peaks observed in September and October.

In the context of inorganic nitrogen fluxes at the sediment–water interface, our results provide an intuitive representation of nitrogen migration and transformation. During most months of the aquaculture period, NO_3_^−^-N was absorbed by the sediment from the overlying water. Remarkably, the abundance of denitrification and anammox genes, namely *narG*, *nirS*, *nirK,* and AMX 16S rRNA, displayed a negative correlation with NO_3_^−^-N flux (Figure 5B). This pattern suggests an important role of nitrification–denitrification and nitrification–anammox processes within the sediment, which may effectively mitigate the accumulation and subsequent release of NO_3_^−^-N.

Furthermore, our results indicate a predominant release of NH_4_^+^-N into the overlying water during aquaculture (Figure 2), probably due to the continuous accumulation of organic matter in the sediment that facilitates ammonification. This process often results in an NH_4_^+^-N production exceeding the processing capabilities of ammonia-oxidizing and anammox bacteria, leading to its release. In addition, the release of NO_2_^−^-N to the overlying water during later stages of aquaculture (Figure 2) suggests limitations in sediment denitrification and anammox processes in managing nitrite levels.

In support of these observations, we observed strong positive correlations between NH_4_^+^-N and NO_2_^−^-N fluxes and the abundance of AOA *amoA*, AOB *amoA*, *nirS*, *nirK,* and *narG* genes (Figure 5). This finding is consistent with the patterns observed in constructed wetland environments by Xu, L. [51], where increased NH_4_^+^-N levels corresponded with higher abundances of denitrification and nitrification functional genes. The high levels of NH_4_^+^-N produced by ammonification, coupled with the NO_2_^−^-N from nitrification, provide rich substrates for denitrification and nitrification processes, thereby promoting microbial growth.

Our study also shows a significant positive correlation between the abundance of the AMX 16S rRNA gene and NH_4_^+^/NO_2_^−^ flux (Figure 5B), highlighting the crucial role of NH_4_^+^-N and NO_2_^−^-N as regulatory factors for anammox bacteria, as previously reported by Zhu, L. [32] and She, Y. [52]. This suggests that the significant accumulation of NH_4_^+^-N and NO_2_^−^-N in sedimentary environments not only influences microbial processes but also provides abundant substrates for the proliferation of anammox bacteria.

### 4.4. Recommendations on Management of the Intensive Hybrid Snakehead Fish Farm

Between July and October, the aquatic environment of intensive hybrid snakehead fish farms experiences a reduced abundance of nitrifying bacteria, which is likely inhibited by sunlight and algae, thus potentially reducing the water’s ability to convert NH_4_^+^-N [36]. The accumulation of organic matter in sediments, under intense mineralization, produces a significant amount of NH_4_^+^-N, which is further enhanced by the bioturbation caused by densely farmed fish, leading to increased release of NH_4_^+^-N into the overlying water. The resulting high levels of NH_4_^+^-N in September and October pose a threat to fish growth and health (Table 3).

To mitigate these potential risks, it is recommended to maintain mechanical aeration during this period to ensure sufficient dissolved oxygen levels in the water, to monitor water quality indicators carefully, and to increase water renewal. The use of mixed nitrifying and denitrifying bacterial preparations during the low light conditions of the evening could also enhance the nitrification capacity of the water body [53]. In addition, setting production limits is crucial for the sustainable development of these high-density fish farms [54]. Setting production limits will help to prevent excessive nitrogen loading, reduce SPM flux resulting from bioturbation, and avoid high levels of toxic nitrogen compounds.

## 5. Conclusions

This research focuses on the study of nitrogen dynamics and the abundance of nitrogen-cycling genes at different stages in hybrid snakehead fishponds. During the culture period, peaks in NH_4_^+^-N and NO_2_^−^-N were observed in September, while Chl-a and NO_3_^−^-N showed a continuous upward trend. An increase in fish biomass significantly enhanced the SPM flux, resulting in the accumulation of TN in both the water and sediment. The sediment mainly acted as a sink for NO_3_^−^-N and, in the later stages of cultivation, as a source for NH_4_^+^-N and NO_2_^−^-N. The SPM and algae in the water, which provide a substrate for coupled nitrification–denitrification processes, together with the abundant presence of denitrifying and anammox bacteria in the sediment, collectively played an important role in both reducing the accumulation of NH_4_^+^-N and NO_2_^−^-N and minimizing their release, thereby slowing overall nitrogen accumulation.

In summary, high-density fishponds are characterized by a significant influx of organic matter and limited water renewal. This often results in intense mineralization at the sediment–water interface and elevated levels of toxic nitrogen compounds in later culture stages. In this context, it is advisable to explore and implement potential strategies aimed at enhancing the management of high-density fishponds and improving their ecological and chemical conditions. However, this study did not focus on nitrification and denitrification rates or changes in the microbial community structure involved in the nitrogen cycle. Further investigation of the structure of the nitrogen cycle is needed to better understand the nitrogen-cycling processes in high-density aquaculture ponds.

## Figures and Tables

**Figure 1 microorganisms-12-00266-f001:**
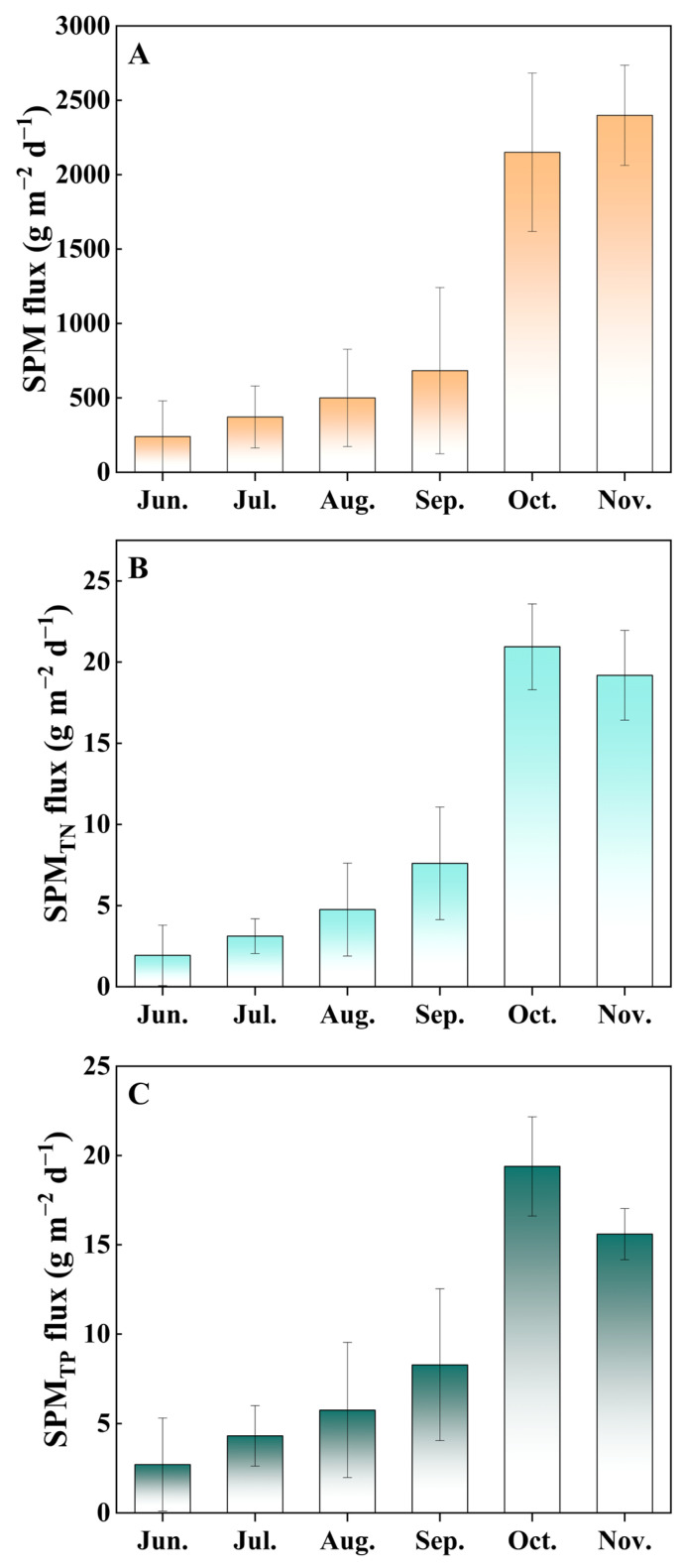
The SPM flux (**A**), TN_SPM_ flux (**B**), and TP_SPM_ flux (**C**) in the hybrid snakehead fishponds during the sampling period. Values are expressed as means with standard deviation.

**Figure 2 microorganisms-12-00266-f002:**
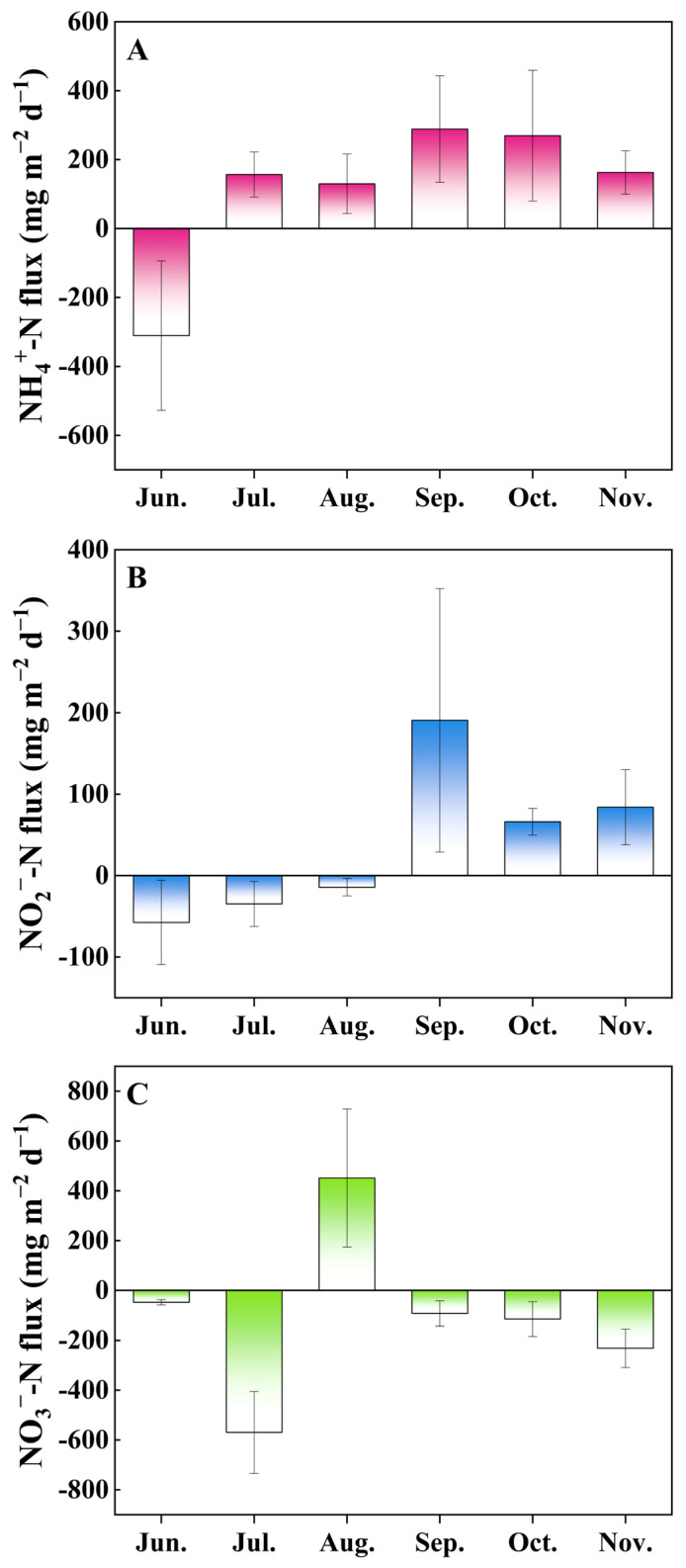
Inorganic nitrogen fluxes of NH_4_^+^-N (**A**), NO_2_^−^-N (**B**), and NO_3_^−^-N (**C**) in the sediment–water interface in the hybrid snakehead ponds. Values are expressed as mean with standard deviation. Positive fluxes indicate nutrient migration from the sediment to the water column, and negative fluxes indicate the opposite.

**Figure 3 microorganisms-12-00266-f003:**
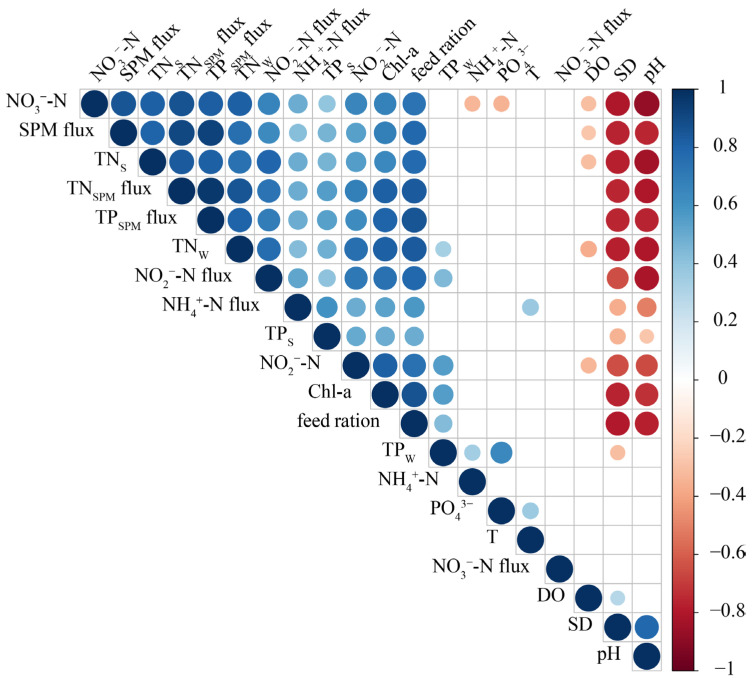
Spearman’s correlations between water column and sediment environmental factors in the hybrid snakehead ponds. Relationships shown are significant (*p* < 0.05). Circle size represents the correlation coefficient, with blue circles indicating a positive relationship and red circles indicating a negative relationship.

**Figure 4 microorganisms-12-00266-f004:**
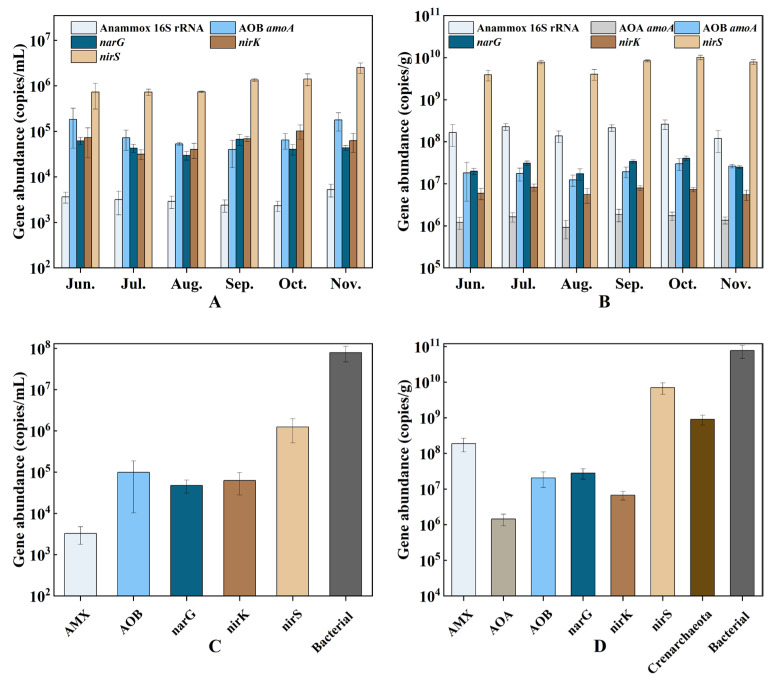
The abundance of nitrogen cycle genes in water (**A**) and sediment (**B**) at different culture stages, and the average abundance of Crenarchaeota, bacterial 16S rRNA, and nitrogen-cycling genes in water (**C**) and sediment (**D**). Error bars indicate the standard deviation.

**Figure 5 microorganisms-12-00266-f005:**
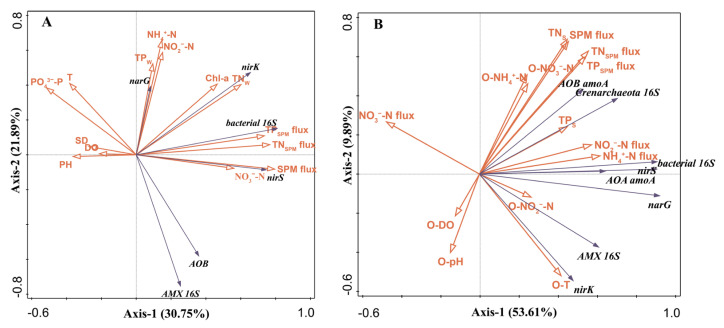
Redundancy analysis (RDA) shows the relationships between gene abundances and environmental variables of the water column (**A**) and sediment (**B**) determined in the hybrid snakehead ponds.

**Table 1 microorganisms-12-00266-t001:** The primer and amplification information of function genes in this study.

Functional Gene	Primer Name	Sequence (5′ to 3′)	Thermal Profile
AOA *amoA*	Arch-amoAF	STAATGGTCTGGCTTAGACG	95 °C for 5 min × 1 cycle; 95 °C for 15 s, 60 °C for 30 s, with a touchdown of −1 °C by cycle, 72 °C for 40 s × 6 cycles; 95 °C for 15 s, 55 °C for 30 s, 72 °C for 40 s × 40 cycles.
	Arch-amoAR	GCGGCCATCCATCTGTATGT	
AOB *amoA*	amoA1F	GGGGTTTCTACTGGTGGT	95 °C for 5 min × 1 cycle; 95 °C for 15 s, 54 °C for 30 s, 72 °C for 40 s × 40 cycles.
	amoA2R	CCCCTCKGSAAAGCCTTCTTC	
*narG*	narG 1960m2f	TAYGTSGGGCAGGARAAACTG	95 °C for 5 min × 1 cycle; 95 °C for 15 s, 63 °C for 30 s, with a touchdown of −1 °C by cycle, 72 °C for 40 s × 6 cycles; 95 °C for 15 s, 58 °C for 30 s, 72 °C for 40 s × 35 cycles.
	narG 2050m2r	CGTAGAAGAAGCTGGTGCTGTT	
*nirS*	nirS1F	CCTAYTGGCCGCCRCART	95 °C for 5 min × 1 cycle; 95 °C for 15 s, 64 °C for 50 s × 40 cycles.
	nirS3R	GCCGCCGTCRTGVAGGAA	
*nirK*	F1aCu	ATCATGGTSCTGCCGCG	95 °C for 5 min × 1 cycle; 95 °C for 15 s, 63 °C for 30 s, with a touchdown of −1 °C by cycle, 72 °C for 40 s × 6 cycles; 95 °C for 15 s, 58 °C for 30 s, 72 °C for 40 s × 35 cycles.
	R3Cu	GCCTCGATCAGRTTGTGGTT	
anammox bacterial 16S rRNA	AMX808F	ARCYGTAAACGATGGGCACTAA	95 °C for 5 min × 1 cycle; 95 °C for 15 s, 54 °C for 30 s, 72 °C for 30 s × 40 cycles.
	AMX1040R	CAGCCATGCAACACCTGTRATA	
bacterial 16S rRNA	1055f	ATGGCTGTCGTCAGCT	95 °C for 5 min × 1 cycle; 95 °C for 15 s, 54 °C for 30 s, 72 °C for 30 s × 35 cycles.
	1392r	ACGGGCGGTGTGTAC	
Crenarchaeota 16S rRNA	771F	ACGGTGAGGGATGAAAGCT	95 °C for 5 min × 1 cycle; 95 °C for 15 s, 54 °C for 30 s, 72 °C for 30 s × 40 cycles.
	957R	CGGCGTTGACTCCAATTG	

**Table 2 microorganisms-12-00266-t002:** The standard curves of target genes.

Target Gene	Standard Curve	R^2^	Efficiency %
AOA *amoA*	Y = −3.71X + 34.89	0.999	86
AOB *amo*A	Y = −3.48X + 36.66	0.998	94
*narG*	Y = −3.66X + 33.93	0.999	87
*nirS*	Y = −3.68X + 39.63	0.999	87
*nirK*	Y = −3.16X + 30.37	0.997	107
anammox bacterial 16S rRNA	Y = −3.27X + 35.07	0.999	102
bacterial 16S rRNA	Y = −3.34X + 38.55	0.993	99
Crenarchaeota 16S rRNA	Y = −4.09X + 44.36	0.999	76

**Table 3 microorganisms-12-00266-t003:** Physicochemical characteristics of water and sediment, monthly feeding ration, and fish biomass across different sampling periods in the hybrid snakehead fishponds. The average values of ponds in different sampling times are indicated (mean ± SD, *n* = 9).

	June	July	August	September	October	November
pH	8.3 ± 0.2 ^a^	8.1 ± 0.1 ^b^	7.6 ± 0.1 ^c^	7.67 ± 0.2 ^c^	7.4 ± 0.1 ^d^	7.3 ± 0.1 ^d^
DO (mg/L)	4.5 ± 0.8 ^ab^	4.8 ± 1.5 ^a^	4.1 ± 1.8 ^ab^	3.5 ± 1.7 ^ab^	4.5 ± 1.2 ^a^	3.1 ± 1.2 ^b^
T (°C)	28.8 ± 0.7 ^d^	31.6 ± 0.6 ^a^	30.9 ± 0.3 ^b^	30.8 ± 0.5 ^b^	29.5 ± 1 ^c^	29.6 ± 0.6 ^c^
SD (cm)	37 ± 4 ^a^	28 ± 4 ^b^	21 ± 4 ^c^	23 ± 2 ^c^	18 ± 2 ^d^	17 ± 1 ^d^
TN_W_ (mg/L)	10.38 ± 3.53 ^d^	12.78 ± 3.27 ^d^	19.44 ± 10.51 ^c^	30.55 ± 3.82 ^b^	36.94 ± 3.52 ^a^	33.89 ± 7.01 ^ab^
NH_4_^+^-N (mg/L)	6.72 ± 4.65 ^a^	2.64 ± 1.99 ^b^	2.07 ± 1.62 ^b^	9.87 ± 5.79 ^a^	8.28 ± 6.37 ^a^	2.47 ± 1.73 ^b^
NO_2_^−^-N (mg/L)	0.59 ± 0.23 ^c^	0.81 ± 0.25 ^c^	2.61 ± 2.1 ^b^	5.97 ± 2.63 ^a^	2.97 ± 1.1 ^b^	1.74 ± 0.48 ^bc^
NO_3_^−^-N (mg/L)	2.6 ± 1.64 ^e^	6.06 ± 4.56 ^de^	11.92 ± 8.09 ^c^	10.44 ± 3.44 ^cd^	17.76 ± 4.73 ^b^	24.34 ± 5.14 ^a^
TP_W_ (mg/L)	2.7 ± 0.31 ^b^	2.88 ± 0.25 ^b^	3.11 ± 0.36 ^b^	3.94 ± 0.77 ^a^	3.98 ± 0.9 ^a^	2.61 ± 0.34 ^b^
PO_4_^3−^ (mg/L)	2.34 ± 0.2 ^b^	2.34 ± 0.3 ^b^	2.37 ± 0.12 ^b^	2.78 ± 0.66 ^a^	2.37 ± 0.12 ^b^	1.01 ± 0.53 ^c^
Chl-a (mg/m^3^)	50.7 ± 25.12 ^e^	131.07 ± 78.12 ^d^	215.41 ± 61.5 ^c^	277.74 ± 43.92 ^b^	411.73 ± 59.54 ^a^	237.69 ± 42.25 ^bc^
TN_S_ (mg/g)	3.42 ± 0.25 ^e^	3.9 ± 0.24 ^d^	3.96 ± 0.26 ^d^	4.5 ± 0.51 ^c^	5.22 ± 0.24 ^b^	6 ± 0.5 ^a^
TP_S_ (mg/g)	3.99 ± 0.26 ^c^	4.91 ± 0.71 ^bc^	4.07 ± 0.71 ^c^	5.59 ± 0.96 ^ab^	6.08 ± 1.43 ^a^	4.86 ± 1.22 ^bc^
O-pH	7.9 ± 0.1 ^a^	7.9 ± 0.2 ^ab^	7.7 ± 0.1 ^bc^	7.6 ± 0.3 ^c^	7.8 ± 0.08 ^abc^	7.3 ± 0.3 ^d^
O-DO (mg/L)	4.2 ± 1.7 ^a^	2.01 ± 0.3 ^bc^	2.3 ± 1.1 ^bc^	2.9 ± 0.4 ^b^	2.7 ± 0.5 ^b^	1.6 ± 0.2 ^c^
O-T (°C)	29.0 ± 0.4 ^b^	30.8 ± 0.3 ^a^	30.9 ± 0.3 ^a^	31.1 ± 0.1 ^a^	29.3 ± 0.8 ^b^	28.8 ± 0.8 ^b^
O-NH_4_^+^-N (mg/L)	6.51 ± 4.25 ^ab^	1.57 ± 1.22 ^c^	2.72 ± 1.36 ^bc^	9.44 ± 5.74 ^a^	8.13 ± 6.09 ^a^	4.97 ± 5.03 ^abc^
O-NO_2_^−^-N (mg/L)	0.68 ± 0.55 ^c^	0.89 ± 0.18 ^c^	2.57 ± 2.06 ^b^	6.08 ± 2.92 ^a^	2.56 ± 1.46 ^b^	1.72 ± 0.32 ^bc^
O-NO_3_^−^-N (mg/L)	2.45 ± 2.13 ^d^	5.82 ± 3.97 ^cd^	11.06 ± 7.02 ^c^	10.09 ± 3.33 ^c^	17 ± 5.73 ^b^	29.31 ± 7.41 ^a^
feed ration (Kg/month)	3756 ± 1464 ^d^	12,245 ± 2230 ^c^	13,568 ± 3444 ^c^	17,467 ± 3924 ^b^	21,087 ± 2981 ^a^	20,347 ± 2042 ^a^
fish biomass (g/m^3^)	444 ± 252 ^f^	1354 ± 410 ^e^	2286 ± 518 ^d^	3062 ± 883 ^c^	4801 ± 980 ^b^	5973 ± 1149 ^a^

Values are the average standard deviation of results. The different letters in the row indicate a significant difference (*p* < 0.05) based on the analysis of variance. “O-” stands physicochemical characteristics of overlying water samples.

**Table 4 microorganisms-12-00266-t004:** The results of the ridge regression analysis.

	Non-Standardized Coefficients		Standardized Coefficients	*t*	*p*	VIF
	*B*	Standard Errors	*Beta*			
intercept	−150.768	153.733	-	−0.981	0.331	-
feed ration	0.009	0.012	0.061	0.769	0.445	1.198
Chl-a	0.689	0.609	0.090	1.130	0.264	1.209
fish biomass	0.309	0.037	0.665	8.342	0.000 **	1.206
*R* ^2^	0.736					
adjusted *R*^2^	0.720					
*F*	*F* (3,50) = 46.527, *p* = 0.000					

Dependent variable: SPM flux ** *p* < 0.01.

## Data Availability

Data are contained within the article.

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
