# Peer review of "Relationship between Nitrogen Dynamics and Key Microbial Nitrogen-Cycling Genes in an Intensive Freshwater Aquaculture Pond"

_microorganisms, 2024, doi:10.3390/microorganisms12020266_

Round 1
Reviewer 1 Report
Comments and Suggestions for Authors
This study focused on investigating nitrogen dynamics and nitrogen cycling genes abundance at different stages in hybrid snakehead ponds. The title matches the contents of the work and is quite detailed. The introduction allows us to assess the current state of the problem at the global level. The “material and methods” of the work are written correctly and allow us to see that all possible sources of error have been eliminated in the study. In terms of level of detail, this section corresponds to articles on similar topics in the world's leading journals. The “Results” section does not contain interpretation of the data, value judgments, elements of the description of the methodology, or references to the literature - only a statement of facts. The article contains the optimal amount of graphic material for this type of research.Overall, the study is well designed and the paper is well-written, but I do have some minor questions and comments that could improve the manuscript:
- The introduction states that this study aims to identify the key factors that may influence the management of fish farms and to propose feasible recommendations to improve the aquatic environment of such fishponds. However, the work does not further discuss how the results obtained will help in the management of fish farms and there are no practical recommendations.
- Table 3. I'm not sure it's necessary to specify values with high precision here. For example, measuring water transparency with a Secchi disk cannot be done with an accuracy of hundredths of cm. Or "feed ration" and "fish biomass" can be specified in integer values. Authors can significantly reduce the number of decimal places for most indicators.
Author Response
Thank you for dedicating your time to assess the manuscript. Below, you'll discover comprehensive responses, along with the highlighted revisions and corrections in the resubmitted files.
Please see the attachment.

Reviewer 2 Report
Comments and Suggestions for Authors
The manuscript entitled “Relationship between nitrogen dynamics and key microbial nitrogen cycling genes in an intensive freshwater aquaculture pond” is aims to biogenic nitrogen migration and transformation in freshwater fishponds during different cultivation periods. The authors discussed the main biochemical pathways of nitrogen, taking into account the processes of its oxidation and reduction, as well as the development of phytoplankton. These findings provide a scientific basis for nitrogen control in hybrid snakehead fishponds and opens up prospects for using RAS systems in such conditions. this study aims to propose feasible recommendations to improve the aquatic environment of such fishponds, so it has important practical significance. To my mind this manuscript is topical and corresponding to the aims and scopes of the “Microorganism’ journal
The introduction shows the most important research results in recent years, which are well systematized by the authorsThe results are presented clearly, statistical methods of data analysis are used.
Here are the comments I found while reading the manuscript
L 63-64 paraphrase the sentence
L 135 YSI water quality meter add url and characteristics: electrodes etc
L 138 142 it is worth writing about the equipment on which the analyzes were carried out
Table 3. It is worth presenting the redox potential data over time, which is one of the key factors in the activation of microorganisms in various parts of the nitrogen cycle. What is the reason for the double accumulation of chlorophyll in October? Did the temperature change on a small number of sunny days? Metabolites of fish culture?
In the discussion, it is worth using data on the redox potential of samples, due to the activation of various microbial processes.
It would be interesting to look at the sample composition based on 16S rRNA gene analysis data.
In conclusion, it is worth presenting in more detail the practical conclusions of the study.
Comments on the Quality of English LanguageMinor editing of English language required
Author Response
We extend our gratitude for your time invested in reviewing the manuscript. Please refer to the detailed responses below, where you will also find highlighted revisions and corrections in the resubmitted files.
Please see the attachment.

Reviewer 3 Report
Comments and Suggestions for Authors
This is a well-described study in which the authors investigated nitrogen migration and transformation in three fishponds during six months (June-November). They focused on the spatial patterns of nutrient concentrations in the ponds, the quantification of nitrogen cycling genes in water and sediment, and the assessment of the relationships between the abundance of nitrogen cycling genes in water and sediment and nutrient concentrations.
There are few comments on the manuscript:
Line 109: when you say „both sets of samples“, do you mean one unfiltered and one filtered? Are the samples for molecular analyses and Chl a taken from the unfiltered sample?
Table 1: I suggest writing the column „Thermal profile“ in a way that makes it clearer which profile is associated with which functional gene
Figure 3: When part of the text is hyphenated (NH4+-N flux, NO2--N flux), there are differences in the length of the hyphen in Figure 3. Please standardise these (make it uniform)
Line 271: Please insert a space between „4.0“ and „x“
Lines 285 and 297: „the main environmental factors“. Please describe how you determined which are the main environmental factors. Have you tested the significance?
Author Response
Your effort in reviewing this manuscript is greatly appreciated. Detailed responses, as well as corresponding revisions and corrections, can be found below and are highlighted in the resubmitted files.
Please see the attachment.

Round 2
Reviewer 2 Report
Comments and Suggestions for Authors
In the revised version of the manuscript, the authors responded to my main comments and significantly improved it. The missing data on the redox potential can be partially replaced by the values of dissolved oxygen, but the authors in future works must take this into account. Data on the analysis of microbial diversity in general are not necessary, although they are very interesting. I believe the manuscript can be published in this form.